# Cloning and Functional Analysis of the *VfRR17* Gene from tung tree (*Vernicia fordii*)

**DOI:** 10.3390/plants12132474

**Published:** 2023-06-28

**Authors:** Li-Yu Liao, Zhang-Qi He, Lin Zhang

**Affiliations:** 1Key Laboratory of Cultivation and Protection for Non-Wood Forest Trees, Ministry of Education, Central South University of Forestry and Technology, Changsha 410004, China; m15367911314@163.com (L.-Y.L.); lshezhangqi@163.com (Z.-Q.H.); 2Key Lab of Non-Wood Forest Products of State Forestry and Grassland Administration, Central South University of Forestry and Technology, Changsha 410004, China

**Keywords:** *Vernicia fordii*, flower development, *VfRR17*, 6-BA, transgene, functional validation

## Abstract

Tung tree (*Vernicia fordii*) is one of the four major woody oilseed species in China. However, its fruit yield is severely hampered by the low number of female flowers and the imbalanced male-to-female flower ratio, which is a problem that restricts the development of the oilseed industry. Previous research has demonstrated that the exogenous application of cytokinins can significantly augment the number of female flowers, although the underlying regulatory mechanism remains elusive. To elucidate the involvement of *VfRR17*, a member of the A-type ARRs family, in the exogenous cytokinin regulation of flower sexual differentiation in tung tree, this study conducted a comprehensive bioinformatic analysis of the physicochemical properties, structural characteristics, and evolutionary relationships of the protein encoded by *VfRR17*. Additionally, gene function analysis was performed using subcellular localization, qRT-PCR, and genetic transformation techniques. The findings revealed that the *VfRR17* gene’s coding region spanned 471 bp, encoding an unstable protein of 156 amino acids with a relative molecular mass of 17.4 kDa. Phylogenetic analysis demonstrated a higher similarity between *VfRR17* of the tung tree and the RR17 gene of *Jatropha curcas*, *Ricinus communis*, *Hevea brasiliensis*, and other species within the Euphorbiaceae family compared to other species, with the greatest similarity of 86% observed with the RR17 gene of *Jatropha curcas*. The qRT-PCR analysis indicated that *VfRR17* exhibited high expression levels during the early stage of tung tree inflorescence buds following 6-BA treatment, peaking at 24 h and displaying a 3.47-fold increase compared to that at 0 h. In female and male flowers of the tung tree, the expression in female flowers during the 1 DBF period was significantly higher than in male flowers, exhibiting a difference of approximately 47.91-fold. Furthermore, notable differential expression was observed in the root, leaf, and petiole segments of the tung tree under low-temperature stress at the 12-h time point. In transgenic Arabidopsis, the *VfRR17* lines and wild-type lines exhibited significantly different flowering times under an exogenous 6-BA treatment at a concentration of 2 mg/L, with the *VfRR17* lines experiencing an 11-day delay compared to the wild-type lines. Additionally, the number of fruit pods in *VfRR17* transgenic Arabidopsis lines was significantly reduced by 28 compared to the wild-type lines at a 6-BA concentration of 3 mg/L. These findings suggest that *VfRR17* likely plays a critical role in regulating flower development in response to exogenous 6-BA, providing valuable insights into the mechanisms underlying exogenous 6-BA-mediated regulation of female flower development in the tung tree.

## 1. Introduction

Cytokinins (CKs) are vital plant hormones that play crucial roles in plant development, including floral bud differentiation, sex differentiation, fruit development, and response to low-temperature stress. Notably, certain exogenous CKs such as kinetin (KT), zeatin (ZT), 6-benzylaminopurine (6-BA), and 2-chloro-4-methylpyridine have been found to impact flower development in different plant species significantly. For instance, in *Prunus avium*, the 6-BA treatment led to increased female flower buds during bud differentiation [1]. In *Castanea henryi*, 2-chloro-4-methylpyridine influenced the sex differentiation of cone chestnut flowers by converting male inflorescences into female ones [2]. Similarly, 6-BA induced the sprouting of carpel primordia on stamen primordia and the early development of inner pistil primordia in *Plukenetia volubilis* [3]. In Brassica napus, 6-BA enhanced flower and ovule production in pistils [4].

Response regulators (RRs) are crucial genes involved in the cytokinin signaling pathway. In Arabidopsis, Arabidopsis Response Regulators (ARRs), categorized into three types (A, B, and C), play significant roles in cytokinin-mediated gene regulation. A-type RRs have a conserved aspartic acid-aspartic acid-lysine (D-D-K) structural domain and a short C-terminus of as yet unclear role, and class A RRs can negatively regulate their transcription [5]. These A-type ARRs act as negative regulators of cytokinin responses, whereas the B-type ARRs function as transcription factors that positively mediate cytokinin-regulated gene expression [6]. Loss-of-function mutations in certain A-type ARR genes in Arabidopsis, such as arr3, arr4, arr5, arr6, arr8, and arr9, have been found to increase the induction of cytokinin major response genes [7]. *ARR4*, *ARR8*, and *ARR9* exhibit relatively high expression levels [8]. On the other hand, overexpression of *ARR7* represses type-A ARR transcripts [9], while ARR15 overexpression enhances cytokinin sensitivity [10]. Additionally, in Arabidopsis, the regulatory network governing stem cell fate in stem tip meristematic tissue involves the CLAVATA (CLV) ligand-receptor system and the WUSCHEL (WUS) 2 protein, which is a positive regulator of stem cells that directly represses transcription of *ARR5*, *ARR6*, *ARR7*, and *ARR15* genes [11]. These findings collectively support the notion that A-type ARRs act as negative regulators in the primary response to CK.

One crucial member of the A-type ARRs family is *ARR17*, which can negatively regulate its transcription. A-type ARRs typically work in conjunction with ethylene to negatively regulate plant growth environment [12]. They have been shown to exert significant regulatory effects on primary root elongation and lateral root growth [13]. Moreover, A-type ARRs have been implicated in mediating low-temperature stress without CKs [14]. In *Populus tremula*, a single gene, *ARR17*, has been demonstrated to control the sexual differentiation, growth, and development of male and female flowers [15]. These studies collectively highlight the considerable influence of the ARR family on plant responses to CKs, with *ARR17* playing a critical role in sexual differentiation in monoecious plants.

The tung tree belongs to the Euphorbiaceae family and is an endemic industrial oil tree in China [16]. Tung trees are unisexual, and their flowering period typically occurs between March and April, yearly [17]. The development of male and female flowers in tung trees involves 12 distinct stages, with the 7th stage marking the initial stage of sex differentiation. During this stage, male flowers of tung trees maintain the parthenosexual state throughout the process of sexual differentiation, while female flowers first go through six stages of bisexual development before the male structures within them deteriorate, resulting in the formation of pure female parthenosexual flowers [18]. Female flowers primarily occupy the central region of the main inflorescence axis and the top of the secondary inflorescence axis, while male flowers surround them, making natural pollination challenging [18]. Furthermore, the low male-to-female flower ratio (approximately 1:27) significantly contributes to a low fruit-setting rate [17]. Further analysis of plant hormone levels and co-expression networks has revealed that the degeneration of male organs in female flowers is accompanied by the accumulation of salicylic acid (SA), which, in conjunction with SA-related genes, co-regulates the programmed cell death of male structures to inhibit stamen development in female flowers. In a study by Liu et al., the application of 6-BA, a plant growth regulator, to male tung trees was found to promote the transformation of male flowers into female flowers. By examining morphological structures and bud differentiation patterns between pure male flowers and monoecious tung trees, the study identified 6-BA as a plant growth hormone that promotes the differentiation of female flowers in tung trees. Notably, a concentration of 640 mg/L of 6-BA resulted in the transformation of pure male flower buds into pure female flower buds, subsequently producing normal fruit. Previous research on Genus Actinidia demonstrated that CK response regulators of the type-C RR family could affect pistil development by regulating carpel development or disrupting stigma structure [19]. Given these findings, the present study aims to investigate the regulatory role and expression function of *ARR17* in the growth and development of tung trees. For this purpose, flower buds of ‘Putao tung’ at various developmental stages were treated with 6-BA, and the *VfRR17* gene fragment was obtained using the GateWay cloning method based on the genomic data of tung trees. Subcellular localization analysis of the *VfRR17* gene was performed, and the gene was introduced into Arabidopsis and tobacco for transgenic functional studies. These experiments provide valuable genetic resources for elucidating the molecular mechanisms underlying female flower development in tung trees.

## 2. Results

### 2.1. Cloning and Sequence Analysis of the VfRR17

Flower buds of the tung tree treated with 6-BA were used as material for total RNA extraction from tung tree flowers (Figure 1A). Reverse transcription was performed, and a target fragment of 471 bp in length (Figure 1B), encoding 156 amino acids, was obtained through PCR amplification using flower bud cDNA as a template and specific primers. The positive recombinant plasmid was sequenced and compared with the total gene library of the tung tree. The sequencing results perfectly matched the obtained sequences and the total gene library. The cDNA sequence of *VfRR17* was successfully cloned and confirmed to be correct.

### 2.2. Structural Analysis of the Protein Encoded by the VfRR17

Protein prediction analysis (Figure 1C) revealed that the protein has a molecular formula of C_752_H_1253_N_207_O_236_S_14_, consisting of 2462 atoms, with a relative molecular mass of 17,419.31 and an isoelectric point of 6.82. The protein comprises 12.8% isoleucine (Leu) and 11.5% lysine (Lys), with no tryptophan residues. Of the 23 negatively charged amino acids (Asp + Glu), they accounted for 14.7%, while the 26 positively charged amino acids (Arg + Lys + His) accounted for 16.6%. Therefore, the protein was positively charged and acidic. The instability coefficient of the protein was 40.01, indicating its instability, while the lipid coefficient was 93.01. The hydrophilicity and hydrophobicity of the *VfRR17* protein were predicted (Figure 1D). The average coefficient of hydrophilicity (GRAVY) was −0.366. The highest hydrophobicity at position 109 was 1.822, and the lowest hydrophilicity at position 71 was −2.844. It suggests that the protein is predominantly hydrophilic with some hydrophobicity. Prediction of the signal peptide of the *VfRR17* gene protein, according to the online website SignalP 5.0, indicated that the protein does not have a specific signal peptide enzymatic cleavage site and is a non-secretory protein. The transmembrane structural domain of the experimentally obtained *VfRR17* protein sequence was analyzed and predicted using TMHMM 2.0 online analysis software. The result did not show the characteristic of having a transmembrane structural domain, and the number of transmembrane helices present was determined to be 0. It indicates that the protein is conserved and lacks a corresponding membrane structure. The secondary structure of the protein encoded by *VfRR17* was predicted using the online software SOPMA 2.0, which showed that out of the 156 amino acids contained in the *VfRR17* protein (Figure 1E), the α-helix accounted for 41.03%, the β-turn angle for 14.10%, the irregularly coiled-coil for 36.54%, and the additional extended chain for 8.97%. It suggests that the α-helix is the major secondary structural element in the protein.

### 2.3. Analysis of Conserved Structural Domains and Phosphosite Prediction of VfRR17 Protein

The amino acid sequences of the *VfRR17* gene and RR17 genes from *Jatropha curcas*, *Ricinus communis*, *Hevea brasiliensis*, and *Manihot esculenta*, along with the *ARR17* gene from *Arabidopsis thaliana*, were compared, revealing an overall similarity of 65.69%, with sequence similarity mainly concentrated at amino acids 25 to 145 (Figure 2A). The sequence similarity was primarily concentrated in amino acids 25 to 145 (Figure 2A). To identify the conserved functional domains of *VfRR17*, online protein structural domain analysis on the NCBI website was performed. Additionally, NetPhos 3.1 software predicted potential phosphorylation sites in the amino acid sequence encoded by the *VfRR17* gene. The analysis identified ten possible phosphorylation sites, including seven serine phosphorylation sites, two threonine phosphorylation sites, and one tyrosine phosphorylation site (Figure 2A). Furthermore, MEME and Pfam online software revealed the presence of motifs 1, 2, and 3 in all species, which are characteristic structural domains of the Response Regulators family. Motif 4 was absent in *AtRR17* of Arabidopsis, while motifs 6–10 were present in *MeRR17X1* of cassava (Figure 2B).

### 2.4. Phylogenetic Relationships of VfRR17 Proteins

To explore the evolutionary relationship between VfRR17 and RR17 proteins in other species of the tung tree, a homologous protein search for VfRR17 was conducted using NCBI BLASTP, resulting in 55 homologous sequences from 37 species. The sequence homology between VfRR17 and RR17 proteins of other plants was compared using MEGA 7 software. A phylogenetic tree was constructed with the Bootstrap parameter set to 1000, and iTOL was employed for visualization and refinement. The results demonstrated that RR17 proteins could be grouped into six species of trees: groups I, II, III, IV, V, and VI (Figure 3). Except for the two species in group VI, the number of species in the other groups did not exhibit significant differences, indicating that the protein sequence of RR17 remained relatively conserved during plant evolution. VfRR17 belonged to group II, sharing the same branch as *Jatropha curcas*, *Ricinus communis*, and *Manihot esculenta*. The sequence similarity between VfRR17 and *Jatropha curca* RR17 was the highest, reaching 86%.

### 2.5. Expression Pattern and Subcellular Localization of VfRR17

In tung tree inflorescence buds treated with 6-BA, with 0 as the control, *VfRR17* exhibited highly significant differences at 12 h, 24 h, 48 h, 3 d, 15 d, 30 d, 60 d, 90 d, 120 d, and 150 d after treatment. The highest expression level was observed at 24 h, which was 3.47 times higher than at 0 h. The expression of *VfRR17* significantly declined at 3 d of inflorescence bud, indicating that the *VfRR17* gene primarily responds to cell division in the early stage following the 6-BA treatment (Figure 4A). During the development of male and female flowers in the tung tree, with 30DBF as the control, the expression of *VfRR17* in female flowers showed an overall increasing trend. The highest expression was observed at 1 DBF, which was 5.63 times higher than that at 30 DBF. In male flowers, the highest expression occurred at 20 DBF, which was 2.01 times higher than that at 30 DBF (Figure 4B). These findings suggest that *VfRR17* may regulate the flowering and maturation of female flowers in the tung tree. Under low-temperature stress in tung tree seedlings, with 0 h as the control, the expression of *VfRR17* in petioles initially increased and then decreased, reaching its peak at 8 h after treatment, which was 5.34-fold higher than that at 0 h (Figure 4C). In the root and leaf parts, the expression at 144 h was 0.2-fold and 0.06-fold at 0 h, respectively, indicating significantly lower expression levels. It indicates that *VfRR17* may play a crucial role in the 4 °C low-temperature stress response of tung tree seedlings. Observation of tobacco leaves injected with Agrobacterium pEaryleyGate104-*VfRR17* after 48 h of light-protected incubation using laser confocal microscopy revealed that the protein fluorescence of *VfRR17* was localized in the cytoplasm, consistent with the predicted analysis (Figure 4D–G).

### 2.6. VfRR17 Responds to Low Concentrations of 6-BA and Affects Flowering Time, Inflorescence Elongation, and Fruit Pod Numbers

To understand the function of *VfRR17* in transgenic Arabidopsis, the expression level of the *VfRR17* gene was examined in the T3 generation of transgenic Arabidopsis strains using fluorescence qRT-PCR. The results indicated an upregulation in the expression of the *VfRR17* gene in different transgenic Arabidopsis strains compared to the wild type, with strain Line 7 exhibiting particularly higher expression levels (Figure 5A). Furthermore, examination of different *VfRR17* transgenic strains and wild-type (WT) Arabidopsis indicated no significant differences in floral organs and fruit clips (Figure 5B,C).

The wild-type and *VfRR17* transgenic strain Line 7 were selected for further analysis. By spraying gradient concentrations of 6-BA with water as a control, the first flowering time, first fruit pod distance, fruit inflorescence length, and the number of fruit pods were observed and compared between *VfRR17* transgenic and wild-type Arabidopsis (Figure 5D–G). Significance analysis revealed that, regarding the first flowering time, *VfRR17* transgenic plants treated with 2 mg/L 6-BA exhibited a significant delay of 11 days compared to the wild type (Figure 5D). Regarding the first pod distance, *VfRR17* transgenic plants treated with 1 mg/L and 2 mg/L 6-BA showed significant differences from the wild type (Figure 5E). Although the infructescence length of *VfRR17* transgenic plants was slightly shortened compared to the wild type under each treatment, the difference was insignificant. However, the number of pods in *VfRR17* transgenic plants treated with 3 mg/L 6-BA was significantly reduced by 28 compared to the wild type (Figure 5F). These results suggest that *VfRR17* responds to low concentrations of 6-BA, leading to delayed flowering, reduced distance between pods and roots, and a decrease in the number of pods.

## 3. Materials and Methods

### 3.1. Plant Materials

The plant materials used in this study were obtained from 8-year-old ‘Putao tung’ trees grown in Qingping Town, Yongshun County, Hunan Province, China, which is the location of the National Tung Tree Germplasm Conservation Repository.

Female and male flower buds of tung trees were collected at four stages: 30 days before flowering (30 DBF), 20 DBF, 10 DBF, and 1 DBF. The samples were frozen using liquid nitrogen for subsequent analysis. Additionally, male tung trees with undifferentiated inflorescence buds were selected in early July 2021 and treated with 6-BA at a concentration of 640 mg/L. The inflorescence buds were collected at various time points: before treatment (0 h), 3 h, 6 h, 12 h, 24 h, 36 h, 48 h, 3 days, 15 days, 30 days, 50 days, 60 days, 90 days, 120 days, and 150 days after the 6-BA treatment. These samples were frozen and stored in an ultra-low temperature refrigerator at −80 °C. Furthermore, ‘Putao tung’ seeds were sterilized with 0.5% potassium permanganate for 30 min and then stored in sandy soil. The soil was kept moist until the seeds germinated, and the seedlings were cultured in an artificial climate chamber at 28 °C until they reached the four-leaf stage. After breaking the soil, the seedlings were transferred to an artificial low-temperature climate chamber at 4 °C. Root, leaf, and petiole tissues were collected at different times: before treatment (0 h), 2 h, 4 h, 8 h, 12 h, 72 h, and 144 h after the low-temperature treatment. These tissues were also stored in an ultra-low temperature refrigerator at −80 °C for subsequent RNA extraction.

### 3.2. Expression Pattern of VfRR17 in Different Plant Tissues

RNA extraction from various plant tissues was performed using the Polysaccharide Polyphenol Plant RNA Rapid Extraction Kit (gDNA clearance column) (Megan, Guangzhou, China). The first-strand cDNA was synthesized by reverse transcription using the HiScript II 1st Strand cDNA Synthesis Kit from Novozymes (Vazyme, Nanjing, China) and diluted 20-fold for qRT-PCR. The cDNA templates of all tung tree samples were amplified using the SYBR^®^ Green Premix Pro Tap HS qPCR Kit (Accurate Biology, Changsha, China). The samples were amplified separately with the internal reference and target genes. The expression assay was conducted using qRT-PCR, and the relative expression levels were calculated using the 2^−ΔΔCt^ method. The expression of *VfRR17* in inflorescence buds of tung trees at the 0 h period of the 6-BA treatment, the expression in female and male flower buds at 30 DBF, and the expression in root, leaf, and petiole tissues at the 0 h period of low-temperature (4 °C), the 0 °C treatment was set as a control. Duncan’s method was used to calculate the relative expression of the *VfRR17* gene in flower buds at different developmental periods of female and male flowers, inflorescence buds after treatment with 6-BA, and under low-temperature stress. The relative expression of the *VfRR17* gene in different tung tree tissues was also calculated using Duncan’s method, and each period’s samples were replicated three times. GraphPad software was used for significance analysis and data visualization.

### 3.3. Bioinformatics Analysis of the VfRR17

The whole genome data of the tung tree was internally assessed, and gene data from other species were obtained from the NCBI database (https://www.ncbi.nlm.nih.gov/ accessed on 22 July 2022). The candidate sequences of the *VfRR17* gene were validated for structural domains and assessed for completeness using the Pfam 32.0 online software (http://pfam.xfam.org/ accessed on 22 July 2022) and NCBI. Physicochemical properties such as isoelectric point, instability coefficient, and molecular weight of the protein encoded by the tung tree *VfRR17* were predicted and analyzed using the ExPASy 3.0 website (https://web.expasy.org/protparam/ accessed on 22 July 2022). The signal peptide of *VfRR17* was predicted using the SignalP 5.0 online website (https://services.healthtech.dtu.dk/service.php?SignalP-5.0 accessed on 22 July 2022). The protein’s secondary structure was predicted using the SOPMA 3.0 online website (https://npsa-prabi.ibcp.fr/cgi-bin/npsa_automat.pl?page=npsa_sopma.html accessed on 22 July 2022). Transmembrane structural domains were predicted using the TMHMM 2.0 online tool (https://services.healthtech.dtu.dk/service.php?TMHMM-2.0 accessed on 22 July 2022). Homologous protein sequences of VfRR17 in different species were obtained from the NCBI database, and a phylogenetic tree was constructed using the neighbor-joining method in MEGA 7, with a Bootstrap parameter set to 1000. Conserved motifs were analyzed using the MEME 5.5.1 online website (https://meme-suite.org/meme/doc/streme.html?man_type=web accessed on 22 July 2022), and cryptic code models of structural domains between different species were constructed using the Web-Logo3 online analysis to investigate their conservativeness.

### 3.4. Cloning and Analysis of VfRR17

An appropriate amount of 6-BA-treated tung tree flower buds was thoroughly ground in liquid nitrogen, and RNA was extracted following the instructions of the Hi Pure Plant RNA Mini Kit (Majorbio, Shanghai, China). The RNA was then reverse transcribed into cDNA using the Hi Script II QRT Super Mix for qPCR (+g DNA wiper) (Vazyme, Nanjing, China) according to the instructions. Primer 6.0 software was employed to design target gene-specific primers for fluorescent qRT-PCR experiments based on primer design principles: Forward primer: 5′-CAATGGAGAAGGTTTTAGGAGA-3′; dlRR17-R: 5′-TTCTGCTGTGGTGACTTTGC-3′. The cDNA amplification was performed using the qRT-PCR kit (Vazyme, Nanjing, China), and the full-length CDS sequence of the *VfRR17* gene was obtained from the tung tree genome database. Target gene-specific primers were designed using Primer 6.0 software, and specific primers were created according to the *VfRR17* gene sequence: Forward primer: 5′-ATGGGTGGTGGGGATTCTTG-3′; Reverse primer: 5′-TCAGCTTCTGCACTTCATTAAATCA-3′. The total amplification system volume was 25 μL, including 1 μL of cDNA, 1 μL of each upstream and downstream primer, 12.5 μL of Mix (HS) Primer STAR, and 19.5 μL of ultra-pure water. The PCR product was visualized through 1.0% agarose gel electrophoresis, and the target fragments were recovered using a gel recovery kit (Vazyme, Nanjing, China) and subsequently ligated to the pClone007 vector. Following transformation with E. coli, positive monoclonal colonies were selected and subjected to sequencing to obtain the cDNA sequence of *VfRR17*. The recombinant plasmid pCAMBIA1300 vector was double-cleaved with EcoRI and SalI, and the recombinant plasmid pCAMBIA1304 vector was double-cleaved with XbaI and SacI. The PMDC32 vector was then ligated using the GateWay method. The positive recombinants were verified by sequencing, and the recombinant plasmids were subsequently extracted.

### 3.5. Subcellular Localization Analysis of VfRR17

The constructed pEaryleyGate104-*VfRR17* plasmid was transferred into Agrobacterium tumefaciens receptor GV3101 using the freeze-thaw method. It was then activated and coated on LB solid medium containing kanamycin (Kan) and rifampicin (Rif). The culture was incubated at 28 °C for two days in an inverted position, and single colonies were selected for colony PCR. Positive colonies and the endoplasmic reticulum marker CD3959 were inoculated in 4 mL of LB liquid medium and incubated overnight until OD_600_ = 0.8–1.0. Then, 1.5 mL of bacterial solution was collected and centrifuged for 5 min at a low temperature to remove the supernatant. Next, using a sterile needle, 1 mL of permeation buffer was added, resuspended, and injected into tobacco leaves. Subcellular localization was observed under a laser confocal microscope. For DAPI staining, a leaf of approximately 1 cm^2^ in size, taken from within the infested area of the leaf, was placed in a solution with a concentration of 3 μL/mL and stained for 10 min, protected from light. The leaf was then rinsed with sterile water and placed on a slide for microscopic observation.

### 3.6. Heterologous Expression of the VfRR17

Arabidopsis overexpression vectors were constructed using the Gateway Rapid Cloning Kit (Takara, Kusatsu, Japan). Primers containing recombinant homologous arms were designed, and positive transgenic plants were identified using the primers for Chaotropic (Hyg): Hyg-F: 5′-ATGAAAAAGCCTGAACTCACC-3′ and Hyg-R: 5′-AATTGCCGTCAACCAAGCTC-3′. Positive Agrobacterium tumefaciens GV3101 carrying the target gene was transferred into wild-type Col-type Arabidopsis thaliana using the pollen tube passage method. Hygromycin was used to screen positive seedlings up to the T3 generation. After the screening, *VfRR17* transgenic Arabidopsis thaliana homozygous to the T3 generation was obtained. A gradient 6-BA solution (1, 2, 3 mg/L) and water were sprayed every three days at the beginning of inflorescence differentiation, using 9 Arabidopsis thaliana plants for each concentration. This spraying process was repeated three times. Wild-type Arabidopsis thaliana was planted as the negative control, and changes in plant organs were observed. The flowering time was measured in the late mature stage of wild-type transgenic Arabidopsis thaliana. The pod spacing, pod number, and pod length of Arabidopsis thaliana were measured and statistically analyzed after different treatments on the main branch (measuring the distance between the first pod away from the root and the fifth pod upwards). The index of Arabidopsis WT was used as a control, and a T-test was conducted using Spss and origin mapping. Each indicator was replicated three times, and photos and records were taken.

## 4. Discussion

The tung tree, belonging to the Euphorbiaceae family, is a woody plant known for its wide adaptability, short growth and development period, and long lifespan. It holds significant importance as an oil tree species in China. However, the poor fruit set caused by female flower abortion in tung trees hampers natural fruit production. To address this problem, increasing the number of female flowers by promoting female and male suppression becomes crucial. CKs, important phytohormones influencing flower buds and inflorescence development, involve A-type ARRs as positive response factors in the CK signaling pathway [7,20,21]. In Arabidopsis, *ARR17* has been identified as a major determinant of flower development. To investigate the important role of *ARR17* in the growth and sex differentiation of tung trees, this study used bioinformatics to obtain 55 homologous sequences of *VfRR17* from 37 species for the first time. Among them, a systematic analysis of *ARR17* members from various Euphorbiaceae species identified *Jatropha curcas*, *Ricinus communis*, *Manihot esculenta*, *Hevea brasiliensis*, and the tung tree itself as having 1, 1, 5, 1, and 1 *ARR17* homologs, respectively. These members were grouped into six distinct species tree groups, with a notable expansion observed in cassava, primarily in group II. In the whole genome data of the tung tree, *VfRR17* exhibited an open reading frame of 471 bp, similar to the coding length of *ARR17* in Populus grandis [22]. The VfRR17 protein possessed an unstable structure of 156 amino acids, with a theoretical isoelectric point of 6.82 and a molecular mass of 17419.31. These characteristics resembled those found in other Euphorbiaceae species and similarities in protein structure with species from the Euphorbiaceae family and Arabidopsis. The results of multiple comparisons showed that the RR17 proteins in Euphorbiaceae demonstrated the presence of the typical structural domain REC A-type ARR, a downstream response factor of cytokinin in Arabidopsis, and a key node in the regulation of CK signaling [6,23], and functions as a positive regulator of CK signaling [24]. However, studies in *Populus tremula* indicated that the downstream response factor *ARR17*, responsible for sexual differentiation in poplar, did not rely on CK action. Instead, the presence of a single induced mutation disrupted the open reading frame of *ARR17*, resulting in the feminization of female organs [15]. Within the JcRR family, JcRR7, a B-type RR member, was expressed in both female and male flowers, while *JcRR8* was exclusively expressed in male flowers, suggesting their potential role as key genes in regulating sexual differentiation [25]. Furthermore, genes associated with increased flower number in response to CKs were found in the phloem tissue of Inca inflorescences [26]. Through multiple sequence alignments of Euphorbiaceae members, it was observed that the motifs of *VfRR17* were highly conserved. However, within subclass II of the constructed evolutionary tree, cassava members expanded and exhibited unknown motifs in the conserved structural domain. This observation suggested that intensive selection following progressive hybridization between *Manihot esculenta* and its two South American wild subspecies may have led to these variations [27,28].

The expression level of *VfRR17* in the floral buds of tung flowers treated with 6-BA peaked at 24 h. 6-BA has been shown to promote an increase in the number of flower buds in various plants, including cotton (*Gossypium hirsutum*) [29], jatropha [30], and blackberry *(Rubus fruticosus)* [31] and other plants. In a separate study, high-throughput RNA sequencing data were utilized to analyze the apple Histone modification gene family (Malus domestica Histone Modifications, MdHMs) under a 6-BA treatment. The analysis revealed the involvement of MdHMs members in the induction process of flower development [32]. Additionally, 6-BA has been demonstrated to enhance flower production in wheat (*Triticum aestivum*) [33]. It has been established that 6-BA can promote the formation of female flowers. At a concentration of 640 mg/L, pure male flowers transform into pure female flowers, increasing the proportion of female flowers within the inflorescence. Transcription data analysis of inflorescence buds treated with 6-BA identified the inhibition of endogenous cell division through the expression of CK biosynthesis pathway genes and signal transduction pathway genes in male tung tree flower buds, ultimately promoting the formation of pistil primordia. In this experiment, *VfRR17* exhibited high expression in inflorescence buds treated with 6-BA at 24 h, suggesting its significance as a key gene in promoting the sex differentiation of tung flower buds. Notably, the expression level of *VfRR17* was significantly higher in female flowers than in male flowers. Previous studies have demonstrated that the stamen degenerates into the ovary base in monoecious trees, eventually disappearing as the flower buds develop from 20 DBF to 10 DBF. This phenomenon occurs due to the premature degradation of the tapestrum layer, preventing the microspore mother cells from undergoing meiosis and leading to programmed cell death (PCD) of the male structures in the stamens of the stamens in female flowers (SFF) [34]. In the present study, the expression of *VfRR17* was high during the development of 1 DBF in female flowers but low throughout the entire development stage in male flowers. Therefore, it is hypothesized that the expression of *VfRR17* is promoted by forming female organs, indicating its close association with female organ development. The findings of this study suggest that *VfRR17* plays a role in the early development of female flowers, as evidenced by its high expression during the late stages of female flower development.

*VfRR17* plays a crucial regulatory role in response to low-temperature stress and exhibits significantly higher expression levels in tung tree petioles than in the roots and leaves. This finding aligns with experimental results that indicate similar high expression in the petiole parts of *Vitis amurensis*, which are involved in low-temperature stress response through hormonal signaling mediated by CKs [35]. The observed high expression in the petiole site at 8 h to 12 h corresponds to Liu’s study, where tung tree leaves displayed significant wilting during 8 h under low-temperature stress [36]. Exogenous CK treatment has been shown to reduce electrolyte leakage in *Nicotiana tabacum* leaves [37]. In rice, a 6-BA treatment combined with a low temperature has been found to increase peroxidase (POD) activity and superoxide dismutase K6 activity, resulting in delayed chlorophyll degradation and promotion of fruit set [38]. Furthermore, 6-BA has been demonstrated to enhance the induction rate of cold-tolerant Actinidia arguta anther healing tissues [39]. However, the present experiment only examined the response to low-temperature stress in tung tree leaves, roots, and petioles and did not investigate the combined effect of low-temperature stress and the 6-BA treatment. Therefore, further investigation is needed to explore their co-regulatory role.

By observing the phenotype of transgenic Arabidopsis plants, it was observed that the *VfRR17* transgenic strain exhibited a significantly advanced flowering time, reduced spacing between fruit pods, and a decreased fruit pod number. Flowering represents a critical transition from vegetative growth to reproductive growth, and the timing of flowering initiation is crucial for successful reproductive growth. Studies have shown that early treatment with 6-BA induces high expression of type A genes that affect the *MdRR* in apple flowering but not type B genes that affect the *MdRR* in apples (*Malus domestica*). Some B-type genes, such as *MdRRB9* and *MdRRB11*, formed a cluster with *ARR1* and actively contributed to apple flower induction through CK signaling, suggesting that the positive effect of CK on apple flower induction is related to the functional role of B-type *MdRR* and genes and flowering time genes [40]. In Arabidopsis, the application of CK (benzylaminopurine, BAP) on the apical axis of the pistil during early development leads to the proliferation of medial structures and changes in the apical structure, thereby affecting flowering [41]. In *Dendrobium nobile*, the transcriptome analysis of RNA-seq data from nutritional and reproductive tissue samples demonstrated the upregulation of several genes involved in inflorescence differentiation promoted the up-regulation of several key cytokines signaling regulators, including genes encoding B-type RRs. Together with TDZ, they significantly upregulated the expression of some marker genes of the GA signaling pathway, indicating the presence of an important low-temperature CK-GA axis during flowering, indicating that the flowering signaling network of Dendrobium flowers is regulated by CK [42]. In Arabidopsis, high-frequency inflorescence regeneration was observed on an induction medium containing zeatin (ZT) and indole-3-acetic acid (3-IAA), and high expression of the inflorescence-specific gene *TFL1* and meristematic-related genes was detected in callus tissue, suggesting the involvement of CK in the regulatory network related to inflorescence development [43]. Additionally, studies have shown that Arabidopsis mutants lacking the ATP-binding cassette transporter *AtABCG14* display defects in long-distance signal transport and nutrient accumulation in roots when affected by CK anti-zein tZ (trans-Zeatin), resulting in delayed taproot elongation, small inflorescence and rosette, slender stem, and altered vasculature [44]. Moreover, transgenic rice plants overexpressing the *OsRR6* type A RR gene exhibit the floret sequence phenotype, indicating the positive regulation of rice inflorescence meristem activity by CKs [45]. Furthermore, CKs have been discussed in other studies as supplementary hormones involved in dwarf stem formation and the induction of flowering in *Rudbeckia bicolor* [46]. Previous research has demonstrated that a 6-BA treatment delays the flowering time of tung trees by approximately 3–5 days compared to the control group. In this study, the *VfRR17* strain significantly advanced the flowering time after the 6-BA treatment, which aligns with the observed phenotype of tung tree flowers after the 6-BA treatment. Advancement in flowering time extends the flowering period and increases fruit set. It is hypothesized that *VfRR17* and 6-BA play pivotal roles in flower development by promoting the advancement of flowering time.

The number of fruit pods represents a significant economic trait in plants. For instance, *Brassica napus* exhibit an increase in pistils and ovules under BAP treatment, leading to more single-seeded pods during artificial pollination. This observation highlights the promotive effect of CKs on fruit pod development in rape plants [41]. Similarly, studies on soybeans (*Glycine max*) have revealed that the basal region of the raceme exhibits relatively high levels of endogenous CKs in the short term. However, upon application of 6-BA to the raceme, a decrease in the number of flowers and pods is observed [47]. By high-performance liquid chromatography-tandem mass spectrometry (HPLC-MS/MS), 14 endogenous CK kinds were analyzed quantitatively in 3 key stages of yield increase (R4-R6) of soybean reproduction and development. The results showed that the content of endogenous CKs in high-yield soybean varieties increased significantly. Notably, a strong correlation between yield components and trans-zeatin (tZ), the most active form of CK, was detected at the R4 and R5 stages. A similar trend was observed for CIS-zein cZ (cis-Zeatin), indicating a potential role of zeatin isomers in both pod and early seed development [48]. Although several factors can contribute to pod abortion, there is currently no functional study investigating the involvement of *VfRR17* in pistil regulation. Nevertheless, observations in Arabidopsis have revealed that *VfRR17* exhibits a phenotype characterized by shortened pod spacing and reduced pod number, providing evidence for the involvement of *VfRR17* in pistil ovary development.

## Figures and Tables

**Figure 1 plants-12-02474-f001:**
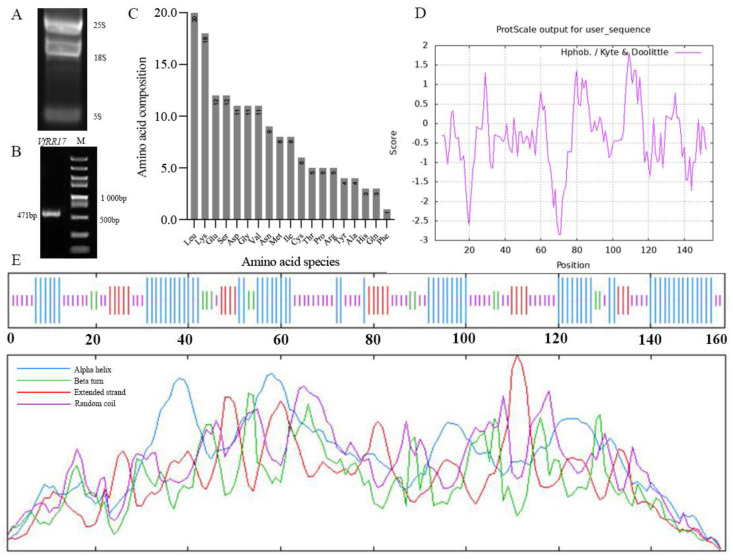
Sequence and amino acid composition of *VfRR17* CDS of the tung tree. (**A**): Electrophoretic map of total RNA extraction from tung tree flower buds treated with 6-BA; (**B**): Detection of *VfRR17* CDS amplification results; (**C**): *VfRR17* protein hydrophilic results; (**D**): Amino acid composition of *VfRR17*; (**E**): Secondary structure of VfRR17 protein.

**Figure 2 plants-12-02474-f002:**
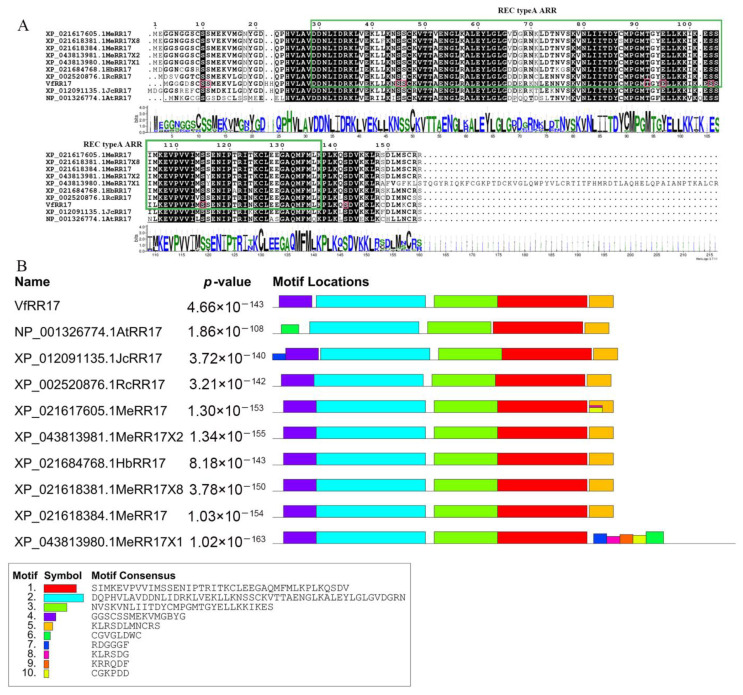
Comparative analysis of multiple sequences of *VfRR17* gene in tung tree and other species. (**A**): Multiple sequence alignment; (**B**): Conserved motifs.

**Figure 3 plants-12-02474-f003:**
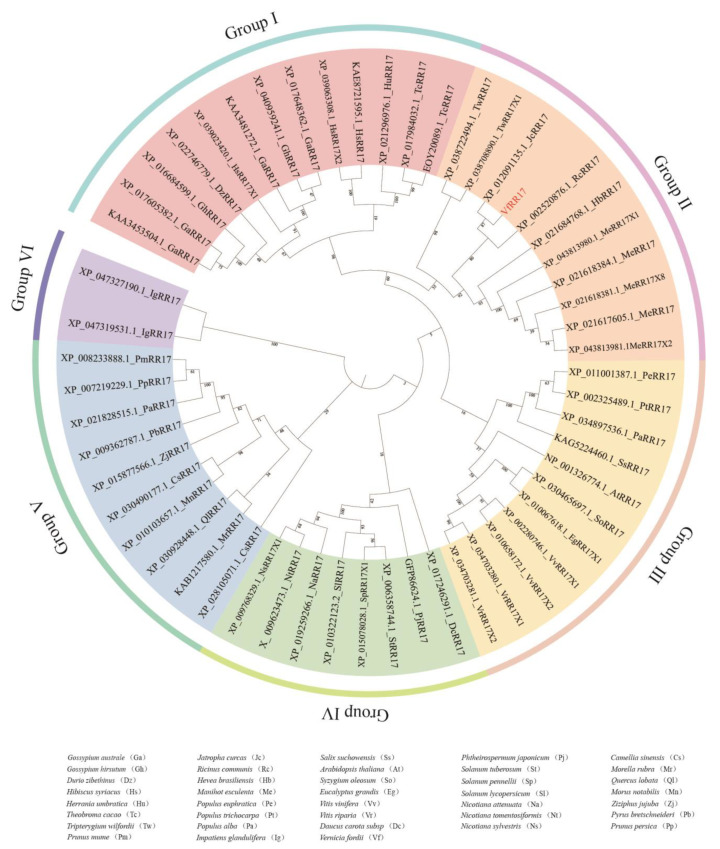
The evolutionary analysis of RR17 protein in tung tree and multi-species.

**Figure 4 plants-12-02474-f004:**
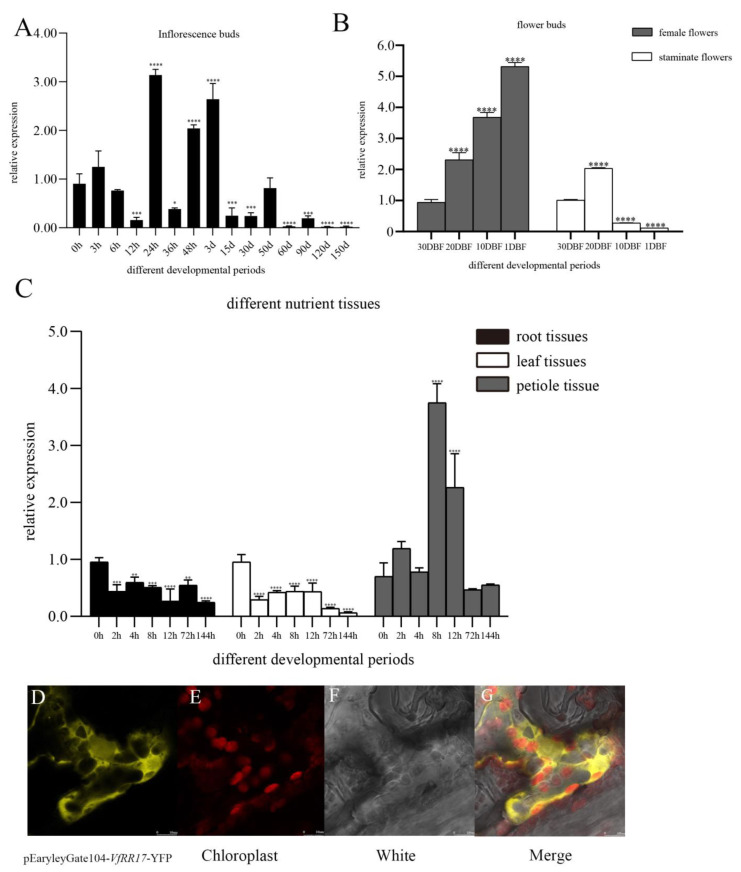
Response pattern analysis and subcellular localization of *VfRR17* gene to 6-BA in tung tree. (**A**): Expression patterns of *VfRR17* on flower buds at different developmental stages without the 6-BA treatment and after the 6-BA treatment; (**B**): Expression patterns of flower bud development in male and female flowers of tung tree in 4 different flowering stages by *VfRR17*; (**C**): Response of *VfRR17* to root, leaf, and petiole parts of tung tree at different periods after low-temperature treatment. Asterisks indicate a significant difference compared to wild-type tobacco (* *p* < 0.05, ** *p* < 0.01, *** *p* < 0.001, **** *p* < 0.0001). (**D**–**G**): represent the yellow fluorescence localization signal, cytoplasmic localization signal, bright field, and superimposed map of *VfRR17* protein, respectively.

**Figure 5 plants-12-02474-f005:**
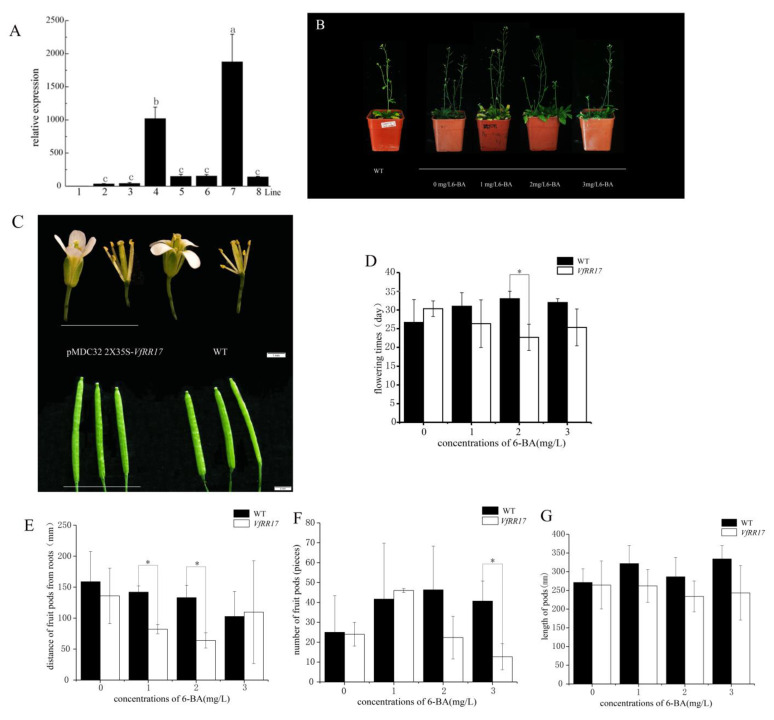
Response pattern analysis of *VfRR17* gene in tung tree. (**A**): qRT-PCR analysis of *VfRR17*-T3 transgenic Arabidopsis Thaliana; (**B**): Morphological comparison of Arabidopsis Thaliana plants in WT and *VfRR17*-T3 generations in response to 6-BA; (**C**): Comparison of flower morphology and fruit pod of *Arabidopsis Thaliana* in WT and *VfRR17*-T3 generations in response to 6-BA; (**D**–**G**) are, respectively: The first flower flowering time, first fruit pod distance, numbers of fruit pods, and the length of fruit pods Arabidopsis thaliana in WT and *VfRR17*-T3 generations in response to 6-BA, respectively. Asterisks indicate a significant difference compared to wild-type Arabidopsis (* *p* < 0.05).

## Data Availability

The samples analyzed may be available upon request after a share transfer agreement. The datasets generated during the current study are available from the corresponding author upon reasonable request.

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
