# Peer review of "Cloning and Functional Analysis of the VfRR17 Gene from tung tree (Vernicia fordii)"

_plants, 2023, doi:10.3390/plants12132474_

Round 1
Reviewer 1 Report
The subject of the submitted manuscript is very important, and as far as can be ascertained at this stage, the authors have carried out a wide-ranging, multi-factor analysis, which bodes well for a good study.
Unfortunately, the work carried out cannot be objectively assessed in this form, because the manuscript is very difficult to understand, and in places not at all understandable. Therefore, I recommend that it be rejected in its current form.
To the authors, I propose the following:
- Re-edit the article, paying attention to accuracy and correcting typos.
- Review the text of the article with an English proofreader (the text is very difficult to understand in its current form)
- The introduction section should be better structured and more literature should be used
- The Materials and Methods section should include a clearer summary of when and how the sample collections were made. For example, it would also be good to know how they knew in advance when the flowering would take place.
- When presenting the results, great attention should be paid to the figures, and an explanation of the symbols should always accompany them.
- In the discussion section, avoid repeating what was said in the introduction section.
I strongly recommend that the article be resubmitted after revision in light of the above considerations.
Pay attention to accuracy and correcting typos.
Review the text of the article with an English proofreader (the text is very difficult to understand in its current form).
Author Response
Dear Reviewer,
We feel great thanks for your professional review work on our article. We apologize for the poor language of our manuscript. We worked on the manuscript for a long time and the repeated addition and removal of sentences and sections obviously led to poor readability. We will have new worked on both language and readability and have also involved native English speakers for language corrections. We really hope that the flow and language level have been substantially improved.
Reviewer 2 Report
Dear Authors,
My review is attached as a pdf file.
Reviewer

Author Response
Response to the reviewer
Dear Reviewer,
We feel great thanks for your professional review work on our article. As you are concerned, there are several problems that need to be addressed. According to your nice suggestions, we have made extensive corrections to our previous draft, the detailed corrections are listed below. The reviewer comments are listed below in italics and specific concerns have been numbered. Our response is given in normal font and changes/additions to the manuscript are given in the blue text.
Major comment:
Comment 1. Lines 255-258: How the RR17 protein was classified into clusters? Which procedure did you use? Which conditions of classification were applied? What was the base establishing significant differences?
Reply:Thanks to the reviewer for the suggestion.To construct the evolutionary tree in VfRR17, we downloaded the homologous protein sequences of VfRR17 in different species from the NCBI database species and used the neighbor joining method in MEGA 7 with the Bootstrap parameter set to 1000 to construct the phylogenetic tree. Our modifications in line140-142,line 262-264.
Comment 2.Lines 274-279: The following sentence is too long and, hence, confusing. “The 274 expression of VfRR17 in the petioles of Tung tree seedlings treated with low temperature stress at 275 4°C first increased and then decreased, with the highest expression at 8 h after treatment being 5.34 276 times that of 0 h (Fig. 4 C); in the root and leaf parts, the expression at 144 h was 0.2 times and 0.06 277 times that of 0 h, respectively, which was significantly lower; thus indicating that VfRR17 may play 278 an important role in Tung tree seedlings treated with low temperature stress at 4°C.” In the here-mentioned sentence, the expression written in bold: significantly lower compared to what and at which probability level? Which statistical procedure did you use when establishing statistically significant differences?
Reply:Thanks to the reviewer for the suggestion. We have made the following changes to your suggestions. To investigate the gene expression pattern of VfRR22 in female and male flowers of tung tree at different developmental stages and in inflorescence buds treated with the exogenous hormone 6-BA and in different tung tree tissues under low temperature stress, significant differences were analyzed using the Duncan method in female and male flower buds of tung tree with 30 DBF as control, in inflorescence buds treated with the exogenous hormone 6-BA with 0 h as control, and in plant tissues subjected to low temperature stress with 0 h as control. Our modifications in line 117-126,276-288.
Comment 3. Lines 295-312: Here my question is the same: line 305: “significance analysis”, line 306: “significantly delayed”: the mention of the statistical background of the analysis is missing;
Line 310: “slightly shorter”: non-significant?
Line 312: “slightly reduced”: non-significant?
Reply: We apologize for our carelessness, which caused your misunderstanding. In line 305, the Fig 5 D~I should be Fig 5 D-G. The data in Fig H and I are irrelevant and we deleted it, but forgot to mark it in the text.
We apologize for any misunderstanding we may have caused you by our poor choice of words. Line 310, there are non-significance between VfRR17 and WT after 2 mg/L 6-BA treatment in number of fruit pods.
Line 312, there are non-significance between VfRR17 and WT after all 6-BA treatment in length of pods.
Once again, We apologize for our carelessness.
Comment 4. Line 320-321: “significant difference compared to 320 wild-type tobacco”: the mention of the statistical background of the analysis is missing;
Reply: We were really sorry for our careless mistakes. Thank you for your reminder. Due to our careless spelling, we mistakenly wrote Arabidopsis instead of tobacco, which we have corrected in line 310-312.
Comment 5. Line 322: “4. Discussion”: I suggest to modify the title of the chapter as “4. Discussion and Conclusions” because this chapter not only compares your results with those of other papers but also summarizes your own results.
Reply: 7. We sincerely thank the reviewer for careful reading. As suggested by the reviewer, we have corrected the “4. Discussion” into “4. Discussion and Conclusions”.
We tried our best to improve the manuscript and made some changes marked in blue in revised paper. We appreciate for reviewer’s warm work earnestly, and hope the correction will meet with approval. Once again, thank you very much for your comments and suggestions.
Best wishes,
Lin Zhang, Prof.
Forestry Ministry, Central South University of Forestry and Technology
410004
No.498 Shaoshan Nan Rd. Changsha
Hunan Province
China
E-mail: triwoodtim918@126.com

Round 2
Reviewer 1 Report
Although there are still some typos, the quality of the manuscript has improved a lot compared to the previous one. The most common typographical error is the absence of spaces between words (most commonly between the word before the reference and the bracket). If you are using MS Word, I recommend that you set the view to highlight spaces so that it is easier to spot these errors. A good solution may also be to run a spell checker for these types of errors.
I suggest a review of the spelling of scientific names. Only the genus name should be written in capital letters and italics should be used.
Additional questions, suggestions:
In line 95 :“The flower buds of female and male flowers of tung tree were collected 30 DBF (30 days before flowering) …” How did you know in advance when the tree would flower? Is it possible to predict the flowering date based on meteorological data for example? The method by which the sampling dates were determined should be accurately described.
Unfortunately, the weakest part of the manuscript is the discussion and conclusions chapter. The chapter is more of a literature review and contains sections that would be more appropriate in the introduction, and no conclusion is drawn. From a grammatical point of view, this chapter also contains the most typing errors. I recommend a thorough review and revision of the chapter sentence by sentence.
The typos are still disturbing in many places, and I recommend a thorough reading and correction of the manuscript from this point of view.
Author Response
Response to the reviewer
Dear Reviewer,
We feel great thanks for your professional review work on our article. As you are concerned, there are several problems that need to be addressed. According to your nice suggestions, we have made extensive corrections to our previous draft, the detailed corrections are listed below. The reviewer comments are listed below in italics and specific concerns have been numbered. Our response is given in normal font and changes/additions to the manuscript are given in the red text and deleted parts are indicated in other colors.
Major comment:
Comment 1.Although there are still some typos, the quality of the manuscript has improved a lot compared to the previous one. The most common typographical error is the absence of spaces between words (most commonly between the word before the reference and the bracket). If you are using MS Word, I recommend that you set the view to highlight spaces so that it is easier to spot these errors. A good solution may also be to run a spell checker for these types of errors.
I suggest a review of the spelling of scientific names. Only the genus name should be written in capital letters and italics should be used.
Reply:We felt sorry for our carelessness. In our resubmitted manuscript, the typo was revised.
Comment 2.In line 95 :“The flower buds of female and male flowers of tung tree were collected 30 DBF (30 days before flowering) …” How did you know in advance when the tree would flower? Is it possible to predict the flowering date based on meteorological data for example? The method by which the sampling dates were determined should be accurately described.
Reply:We were really grateful for your expert evaluation work on our content. Previous studies by Li (17. Li, W., MeilanDong, XiangCao, HepingWu, YeShang, HaiHuang, HuimengZhang, Lin) support this.Structure and function of trees. 2020, 34, doi:doi.org/10.1007/s00468-020-02041-3.), we chose the tung tree's flowering stage for our study of the flower biology and ontogeny.The leaf primordium stage, inflorescence primordium stage, inflorescence axis differentiation stage, and flower organ differentiation stage were the four stages used to categorize the flower bud development of the tung tree. Mid-June to early July is when floral bud distinction was most important. In order to sample when male and female flowers began to distinguish, we started monitoring their growth in mid-June. In line 78–93, we added a literature review as an addition to this section.
Comment 3.Unfortunately, the weakest part of the manuscript is the discussion and conclusions chapter. The chapter is more of a literature review and contains sections that would be more appropriate in the introduction, and no conclusion is drawn. From a grammatical point of view, this chapter also contains the most typing errors. I recommend a thorough review and revision of the chapter sentence by sentence.
Reply:I appreciated your suggestion. The conversation was rewritten to be more reasonable and rational. We made some edits to the manuscript in an effort to make it better. Our changes could be indicated in the whloe article. The structure and substance of the document would not be affected by these changes. We also asked a friend of ours from the United States who speaks native English to help us polish our piece. We also hoped that you will accept the amended document.
We tried our best to improve the manuscript and made some changes marked in red in revised paper. We appreciate for reviewer’s warm work earnestly, and hope the correction will meet with approval. Once again, thank you very much for your comments and suggestions.
Best wishes,
Lin Zhang, Prof.
Forestry Ministry, Central South University of Forestry and Technology
410004
No.498 Shaoshan Nan Rd. Changsha
Hunan Province
China
E-mail: triwoodtim918@126.com

Round 3
Reviewer 1 Report
The authors have put a lot of work into improving their manuscript this time. They have invested the most effort in correcting language errors and discussing their findings. In my opinion, the manuscript in its present form is well understood and the essence of the work has been successfully highlighted. I thank the authors for their thoughtful corrections and I believe the article is ready for publication in its present form.